# Bradykinin and Neurotensin Analogues as Potential Compounds in Colon Cancer Therapy

**DOI:** 10.3390/ijms24119644

**Published:** 2023-06-01

**Authors:** Magdalena Szaryńska, Agata Olejniczak-Kęder, Kamila Podpłońska, Adam Prahl, Emilia Iłowska

**Affiliations:** 1Department of Histology, Faculty of Medicine, Medical University of Gdansk, 80-210 Gdansk, Poland; magdalena.szarynska@gumed.edu.pl (M.S.); agata.olejniczak@gumed.edu.pl (A.O.-K.); kpodplonska@gumed.edu.pl (K.P.); 2Department of Organic Chemistry, Faculty of Chemistry, University of Gdansk, 80-308 Gdansk, Poland; adam.prahl@ug.edu.pl

**Keywords:** bradykinin, neurotensin, colorectal cancer, peptides

## Abstract

Colorectal cancer (CRC) is one of the most lethal malignancies worldwide, so the attempts to find novel therapeutic approaches are necessary. The aim of our study was to analyze how chemical modifications influence physical, chemical, and biological properties of the two peptides, namely, bradykinin (BK) and neurotensin (NT). For this purpose, we used fourteen modified peptides, and their anti-cancers features were analyzed on the HCT116 CRC cell line. Our results confirmed that the spherical mode of a CRC cell line culture better reflects the natural tumour microenvironment. We observed that the size of the colonospheres was markedly reduced following treatment with some BK and NT analogues. The proportion of CD133+ cancer stem cells (CSCs) in colonospheres decreased following incubation with the aforementioned peptides. In our research, we found two groups of these peptides. The first group influenced all the analyzed cellular features, while the second seemed to include the most promising peptides that lowered the count of CD133+ CSCs with parallel substantial reduction in CRC cells viability. These analogues need further analysis to uncover their overall anti-cancer potential.

## 1. Introduction

Colorectal cancer (CRC) is one of the most common cancers globally, accounting for 9.2% of all cancer deaths. It ranks as the second most lethal cancer and the third most prevalent malignant tumour worldwide [1]. According to the Polish National Cancer Registry, in 2018, colorectal cancer was the third most common malignant neoplasm of men in Poland and the second most frequently diagnosed malignant neoplasm in women. In the same year, 18,701 new cases of CRC were diagnosed, and 12,500 deaths due to CRC were registered in Poland, with those numbers continually increasing [2,3]. The WHO noticed over 1.85 million new CRC cases worldwide [4]. Whilst surgery remains an essential and primary option for treatment, in cases where CRC is discovered early, chemotherapy and radiation therapy are necessary for late stages of CRC. As many CRC patients are diagnosed with distant metastases and almost 50% of other patients will develop metastasis during their lifetime [5], there is an enormous need to develop some novel therapeutic protocols which overcome the natural resistance of cancer stem cells (CSCs).

Bradykinin (BK, Arg-Pro-Pro-Gly-Phe-Ser-Pro-Phe-Arg) is a very potent, nonapeptide that acts as an inflammatory mediator. It is a constituent of the kallikrein-kinin system (KKS) [6,7,8,9,10]. Bradykinin activates several second messenger systems in healthy organisms, thereby regulating blood–brain barrier permeability, blood pressure, pain perception, glutamate release from astrocytes, neuronal differentiation, and nitric oxide production [11]. The action of BK is mediated through an interaction with cell surface BK receptors, namely, Bradykinin Receptor B1 (B1R) and Bradykinin Receptor B2 (B2R), which belong to the G-protein-coupled receptor (GPCR) family [6,11]. Both receptors are activated by carboxypeptidase degradation products of BK and kallidin (KD), specifically [des-Arg9]-BK and [des-Arg10]-KD, respectively. B1R has minimal expression in healthy tissues, and its expression is induced only under special conditions such as injury and inflammation [12,13]. In most cases, BK acts through B2R, which is ubiquitously expressed and implicated in vasodilation, osmoregulation, smooth muscle contraction, and nociceptors’ activation [6]. However, B2R expression upregulates under pathological conditions of tissue injury [14,15,16,17,18].

The huge role of these receptors has prompted scientists, including our group, to search for antagonists that would improve the effects of potential therapy in the future. The relationship between the peptide sequences and their activity has been carefully studied [19,20,21,22,23,24,25]. For example, acylation of the *N*-terminus suppresses antagonistic activity [26], yet substitution using a bulky group such as 1-adamantaneacetic acid (Aaa) resulted in a significant increase in B2 antagonistic activity in the rat blood pressure test [27,28]. Substitution of the l-proline residue at position 7 (Pro7) with d-phenylalanine ([d-Phe7]) BK is the key structural change in the sequence of BK that alters the steric position of the C-terminal dipeptide Phe8–Arg9 and leads to antagonistic B2R activity [29,30].

BK receptors are expressed in different human cancers [6], suggesting a potential role in inducing pathologic signal transduction in cancer growth and progression, nitric oxide production, and vascular permeability enhancement in tumours. This peptide seems to be an autocrine growth factor in cancer niches, and thus BK antagonists offer promise for cancer treatment [31,32]. BK receptors are widely expressed on these cancer cells, and stimulation of BK production has been demonstrated [33,34]. BK is a multivalent growth stimulant: it stimulates cancer growth angiogenesis by the release of fibroblast growth factor (FGF) and vascular endothelial growth factor (VEGF). In addition, BK increases tissue permeability to facilitate tumour invasion by membrane metalloproteases (MMPs).

Neurotensin (NT) is a peptide hormone consisting of 13 amino acids with the sequence *p*Glu^1^-Leu-Tyr-Glu-Asn-Lys-Pro-Arg-Arg-Pro-Tyr-Ile-Leu^13^-OH. NT performs its intracellular effect by interacting with four cell surface receptors, namely, NT receptor types 1-4 (NTSR1-4) [35]. NTSR1 and NTSR2 belong to the GPCR family, with high and low affinity toward NT binding, respectively. Although NT is composed of 13 amino acids, the NT(8−13) fragment (Arg^8^-Arg-Pro-Tyr-Ile-Leu^13^) is sufficient to retain the full biological activity of the native peptide [36]. Consequently, most derivatives have been developed from the shorter biologically active NT(8−13) peptide. However, the short in vivo half-life time of NT(8−13) (approximately 2 min) is a major drawback to its use as an active compound.

NT, which is released primarily by ingestion of dietary fats, facilitates intestinal mucosal proliferation, free fatty acid absorption, lipid metabolism, and glucose homeostasis [37,38]. NT stimulates the production of digestive enzymes by the pancreas [39,40]. The well-characterized physiologic effects of NT are primarily endocrine mediated; however, NT is also known to act in a paracrine and autocrine fashion in some cell types, such as certain cancers and intestinal cells [41,42,43]. Importantly, findings have shown that NT contributes to the maintenance of intestinal homeostasis through regulation of extracellular signal-regulated kinase 1/2 (ERK1/2) and WNT/β-catenin signaling. An evolutionarily conserved role for NT in the maintenance of intestinal stem cells (ISCs) during nutritional stress has been identified [44]; however, whether the intestinotrophic effect of NT is mediated through the regulation of ISCs still requires elucidation.

When considering the role of NT in the treatment of colorectal cancer, it has been proven that it stimulates the growth of mouse and human colon cancer cell lines in tissue culture and after their xenograft transplantation into nude mice [45]. Other in vivo studies showed that NT administration stimulates size and weight of a tumour, and significantly stimulated the DNA, RNA, and protein content of MC-26 tumours in mice [46]. NTSR1 expression was significantly correlated with an increased tumour number when sporadic cancer was generated in mouse models. However, no effect on NTSR1 expression was noted on the number of aberrant crypt foci, suggesting that the NT/NTSR1 signaling complex plays an important role in promoting the conversion of precancerous lesions into adenoma [47]. In addition, NT enhances colorectal cancer cell migration by increasinginterleukin-8 (IL-8) expression and secretion. These effects are blocked by NTSR1 antagonists and curcumin [48]. The effect of NT on IL-8/CXCL8 expression may depend on NF-κB activation, as demonstrated by transfected NCM460-line colonocytes and HCT116 human colorectal cancer [48]. Interestingly, the overexpression of sortilin (SORT1)/NTSR3 in prostatic, colonic, and pancreatic cancers showed that sortilin in association with NT is responsible for cancer cell growth [49]. In the colorectal cancer HT29 cell line, sortilin induced cytoskeleton and desmosomes reorganization, which was followed by increased migratory abilities of the malignant cells [50,51].

In our project, we studied the effect of structural modification of BK and NT on their anticancer properties. For this purpose, we used modified peptides, and the biological anti-cancerous features of these were analyzed using two colorectal cancer cell lines originally collected from donors with cancers of different progression statuses. The HCT116 cell line represents CRC with a higher progression status in comparison to HT29 cells (TNM3/Dukes’ D versus TNM2/Dukes’ C, respectively) [52]. In addition, we expanded CRC lines in two distinct forms: adherent (recommended by the supplier) and spherical, which better corresponds to the natural tumour microenvironment and presents properties that may be more relevant to in vivo tumour development than cancer cell lines cultured in adherent forms [53,54]. Such diversification of the cellular forms allowed us to directly compare the usefulness of both forms of CRC cell lines for in vitro investigations. All analyses were conducted in comparison to specific control samples.

## 2. Results

### 2.1. Synthesis and Characterization of the Peptides 

To determine anti-cancer potential in the treatment of colorectal cancer, we chose two neurotransmitter peptides for our study: BK and NT. 

In the case of BK, analogues with conformationally limiting residues at positions 7 and 8 were prepared in order to somewhat force the β-turn of the C-terminal fragment of the peptides. For this purpose, we used compounds such as 1-aminocyclopentyl-1-carboxylic acid (Apc) and 1-aminocyclohexane-1-carboxylic acid (Acc). These modifications were made based on the properties of bradykinin analogues that we have verified. They show that the presence of suitable achiral, non-aromatic, conformationally constrained amino acids, e.g., Acc, at these positions affects antagonistic properties against the B2R receptor [22,23,25]. Another modification that changed natural amino acid (l) to unnatural (d) amino acid residues means that the enzymes “did not recognize” the amino acid in the sequence as natural, and as a result, proteolytic cutting did not occur. In addition, *N*-terminal modifications were also introduced. It has been proven that acetylation of the *N*-terminal of the BK analogues with appropriate reagents affects antagonistic properties against the B2R receptor [25].

In the case of NT, an increase in the stability of this compound seems to be crucial. There are many analogues known in the literature that show strong affinity for the NTRS2 receptor, but which are very quickly degraded by enzymes. As the active site of this compound is located between residues 8 and 13, we made modifications in this fragment. A non-protein l-*tert*-leucine (Tle) residue was introduced, resulting in a significant inhibition of degradation. At the same time, this did not significantly change the nature of the compound, and therefore further specific binding to the receptor was possible [55,56,57]. 

Based on our previous research, we chose the set of BK and NT analogues that are shown in Table 1 and Table 2. For bradykinin, we designed eight analogues numbered BK-1 to BK-8, and for neurotensin, we designed six, numbered 9–14. The synthesis and characterization of the peptides were performed as described in Section 4. All mass spectra and chromatograms of the peptides were included in the Appendix A. During design of the peptides, in the case of BK and its analogues, we used our experience in studying the structure–bioactivity relationship [22,23,24]. The compounds were designed to exhibit the greatest antagonistic potential towards the B2R receptor. In the case of NT, the most important fragment is considered to be the residues 8–13 because this section is responsible for binding to the NTSR1 receptor. All peptide sequences and the aggregate characteristics of the purified products are presented in the tables below.

### 2.2. Stability Test

To check the stability of the studied peptides in human plasma, 24 h incubation of peptides in the appropriate medium was performed. For the studies, two methods were used. The first one involved the presence of trichloroacetic acid (TCA) as a protein precipitating reagent, while the second one involved a mixture of formic acid and urea. The first method was used for peptides BK, BK-1, BK-2, BK-5, BK-7, and NT, as well as peptides NT-9 to NT-14. The procedure with the mixture of the formic acid and urea was used for BK-3, BK-4, BK-6, and BK-8. Two different methods were necessary because TCA caused precipitation of the peptide itself, which disrupted the experiment. Bradykinin as a native peptide and as a control is a relatively stable compound. After 4 h of incubation, no significant degradation was observed (96% of the peptide concentration), while after 24 h in the test sample, about 50% of the peptide remained. A similar situation was observed for the peptides BK-1 and BK7. Slightly higher stability, after 24 hours in the test sample was 70% of the peptide, had analogue BK-8. The most stable were analogues BK-2, BK-4, and BK-5. The last one and the least stable was BK-3. Its final concentration was about 40% of the initial concentration value (Appendix A).

In the case of neurotensin analogues, the native peptide was the least stable. It was successively degraded during the experiment, and after 24 h, it was practically absent from the test sample. All neurotensin analogues (NT-9 to NT-14) had high resistance to enzymatic degradation (Appendix A).

Conducting stability tests is very important because of the potential use of the analyzed compounds in anticancer therapies. It is known that both native peptides produced in the body are unfortunately rapidly degraded by proteases. Therefore, we designed our compounds to potentially increase their resistance to enzymatic degradation.

### 2.3. In Vitro Preliminary Studies

The initial study demonstrated that the BK and NT analogues induced enhanced proliferation of HT29 cells (Figure 1), and these results were not in accordance with our main goal, which was to find BK and NT antagonists with anti-cancerous activity. At the same time, HCT116 cells responded in a different manner to treatment with BK and NT analogues, and thus we decided to continue our experiments with the HCT116 cell line (results concerning proliferation analysis are presented in Figure 1C,D). These preliminary data caused the fact that BK as well as NK action is mainly associated with interaction with specific receptors (B1R, B2R, or NTRS1 and NTRS2); we wanted to see how the expression of these molecules changed in the selected cell lines. We conducted a real-time PCR assay to verify the expression of BK and NT receptors within both cell lines to explain such diversity (Figure 2). We found that the expression of B1R and B2R was four or six times higher in HT29 cells in comparison to HCT116 cells, respectively. With regard to NT receptors (NTSR1 and SORT1), similarly large changes were found, with the expression of these two receptors being 1.5 or 4 times higher, respectively, in HT29 cells than in HCT116 cells. This may explain why both CRC cell lines reacted in different ways to our analogues. The concentration of some membrane receptors may influence or even determine its activity; for instance, the level of Fas in cancer cell membranes required for pro-apoptotic signaling has been estimated to be 1000 times higher than the level necessary for Fas pro-survival activity [58]. Numerous analyses are required to verify this hypothesis in the context of BK and NT receptors. At the same time, observations concerning the impact of culture mode on the expression of BK receptors revealed that it increased 12–14 or 1.5–2 times for HCT116 and HT29 cells, respectively, when we compared spherical cells with adherent counterparts. The results for NT receptors did not show statistical significance.

### 2.4. Analysis of CRC Cell Phenotype with Flow Cytometry

HCT116 cells in adherent (Figure 3) and spherical (Figure 4) form were subjected to cytometric analysis of phenotype following treatment with BK and NT analogues because we wanted to check the effect of the peptides on the growth of CD133+ cells. CD133 is one of the key biomarkers for isolation and characterization of stem cells. Increasing evidence has shown that CD133 is not only a biomarker but that it functions also in cell growth, development, and tumour biology. We evaluated the proportion of CSCs with commonly used CSC-like markers, namely, CD133, CD44, and CD29 [59,60]. The general count of CD133+ CSCs of CRC cell lines cultured in the adherent form were not affected (Figure 3A,B). HCT116 cells were also analyzed with respect to the presence of CD29 and CD44 integrins on their surface. According to our previous observations, the adherent populations of CRC lines are 99% CD133+, whereas spherical cultures are heterogenic, more like CRC cells isolated from tumour fragments [61]. Considering that 99% of HCT116 CRC cells expanded in the adherent form were CD133+ cells as we presented in our previous study [54], the CD44 and CD29 markers were analyzed only within the CD133+ population (Figure 3C,D). We noted that BK analogues prominently increased the numbers of the cells after 24 h incubation, when compared with untreated control cells. Among the NT analogues, only the effect of NT-11 and NT-12 were found to be substantial. 

Cells in colonospheres were found to be more sensitive to our treatment as we revealed some growth fluctuations in the CD133+ cell count (Figure 4A,B). Surprisingly, the proportion of CD133+ cell within the general population increased significantly following incubation with the analogues BK-2 and NT-14. Similarly as for the adherent HTC116 cells, we also tested the spherical form with respect to the presence of CD29 and CD44 integrins on their surface but not only in CD133+ (Figure 4C,D) but also in CD133 (Figure 4E,F). In these studies, the count of CD133+, CD44+, and CD29+ cells was not affected in spherical culture, while at the same time, we observed that CSCs in colonospheres were more likely to change the level of these markers within the CD133-negative population. The analogues BK, BK-1, and BK-2 prominently elevated the CD133-CD44+CD29+ cell count in comparison with untreated control cells. In contrast, the analogues BK-4, BK-6, and BK-8 decreased the number of these cells. The NT-derived peptides exerted a weaker impact, with only the wild form significantly increasing the number of CD133-CD44+CD29+ cells. The NT-14 analogue lowered the number of these cells below the NT-induced level. 

CD44 and CD29 are integrins; therefore, we hypothesized that HCT116 cells under the influence of BK and NT analogues may have improved their adhering abilities in the monolayer and may have strengthened junctions between cells within colonospheres to maintain their integrity.

### 2.5. Colonosphere Formation and Quantification

BK and NT analogues triggered visible changes in the morphology of spheres derived from the HCT116 cell line. The sizes of the colonospheres were significantly reduced when the cells were treated and the number of dead cells and debris in the medium increased following treatment with the analogues (Figure 5). These results could be associated with the protective strategy of cancer cells, which lower their proliferative activity in disadvantageous environments in order to become less sensitive to therapeutic compounds.

### 2.6. Analysis of CRC Cell Viability

We evaluated the percentage of nonviable cells of HTC116 following analogue treatment via a flow cytometric assay using 7-Amino Actinomycin D (7-AAD) dye according to the procedure presented in the Materials and Methods Section 4.6. This dye is excluded by living cells but binds selectively to the DNA of damaged permeabilized cells. We learned that the number of 7AAD-positive cells changed according to different scenarios in CRC cells cultured in adherent and spherical forms following the incubation with BK and NT analogues (Figure 6).

The cells in the colonospheres seemed to be more sensitive to BK and BK–derived peptides, whereas NT and NT analogues exerted milder effects. The BK-derived peptides with Aaa at the 0 position (BK-3, BK-4, BK-6, BK-8) were significantly cytotoxic, and this effect was the same following incubation with BK peptide. NT peptide prominently increased the number of nonviable 7AAD+ cells. Surprisingly, the NT-14 peptide preserved HCT116 cells as the number of dying cells was seen to significantly decrease.

Furthermore, we observed that the viability of adherent HCT116 cells was elevated under the influence of BK, but the BK derived peptides did not change the 7AAD+ cell count when compared with unstimulated control. At the same time, NT exerted a more intense effect by either increasing or decreasing the 7AAD+ cell count. The short NT-derived peptides with added alanine in their chain (NT-10 and NT-11) decreased the number of dying cells within the adherent population. The peptides created by replacement of the ninth amino acid with lysine (NT-12 and NT-14) reduced the viability of cells after incubation. We noticed that these peptides induced opposite effects in the spherical culture. These observations are consistent with results that we obtained using the cytotoxic test (with MTS reagent). The only exception was the influence of BK-7 on the viability of adherent cells, where we noticed opposite results using these two methods. Incubation of colonospheres with the tested analogues resulted in a decreased viability, in a limited manner; however, the percentage of non-living cells was markedly higher in the adherent model of HCT116 in comparison to its spherical counterpart. Our observation was consistent with our previous results confirming that spheres are a more resistant platform toward diverse therapeutic agents [53].

## 3. Discussion

In the present study, we analyzed the effects of two neurotransmitters, BK and NT, and analogues of these as potential therapeutics in colorectal cancer. Analogues with defined antagonistic properties were initially designed and synthesized [20,22,23,24,27,28,62]. For the aforementioned BK analogues, two types of assays were performed in our laboratory: the in vitro rat blood pressure test and the in vitro rat uterus test against the B2R receptor [22,23,24]. Our attention was focused on analogues containing two basic modifications. The first modification involved acetylation of the *N*-terminal fragment of the peptides with 1-adamantaneacetic acid (Aaa)—these are analogues BK-3, BK-4, BK-6, and BK-8. The second modification was an exchange of significant positions relative to Stewart’s antagonist, namely, positions at 7 and 8 with Apc (BK-1, BK-2), Acc (BK-3, BK-4), and Pip (BK analogues 5–8). 

Apc-containing compounds did not significantly change the cancer cells properties, including that they did not influence the count of CDD133+ CSC cells. Other studies of antagonistic properties showed that the BK-2 analogue has strong antagonistic activity not against B1R but against the B2R receptor [23]. Our preliminary PCR analysis of the expression of BK receptor genes revealed that HCT116 CRC cells expressed both receptors at a low level and that this could be the reason for the discrepancy between the results of the antagonistic activity and inhibition of the CSC cells.

The BK-3 peptide showed promising properties in the studies presented in this paper on CRC cell lines, as well as a reduction in the number of CD133+ CSC. The compound has a modification at position 7 in the form of a non-chiral, large cyclic, sterically constrained noncoded amino acid, specifically 1-aminocyclohexane-1-carboxylic acid (Acc). This change reduced the flexibility of the peptides, thus forcing the peptide backbone and side chains to adopt specific orientations, while at the same time limiting conformational freedom. In addition, this compound is acetylated at the *N*-terminus by Aaa. Both of these modifications exerted significant impact on the antagonistic properties, which has been presented by other groups [22,23] and also in our study.

The next two important bradykinin analogues, namely, BK- 6 and BK-8, belong to a broader group of analogues in which two modifications of the primary structure have been made. The previously mentioned, significant positions (7 and 8) have been replaced by l-pipecolic acid and D-pipecolic acid in BK-6 and BK-8, respectively. The second modification was also acetylation on the *N*-terminus by Aaa. Previous studies indicated that these types of analogues possessed strong antagonistic properties against the B2R receptor. It has been demonstrated that analogues containing pipecolinic acid with the L-configuration (identical to the native protein amino acid configuration) possess stronger antagonistic activity than with the D-configuration [24,62]. Both of these peptides have also been tested against U87-MG human glioblastoma cells or human lung carcinoma cells and also inhibit the growth of the cancer cell lines, mostly by the B1R receptor [30,63,64]. It is well known that this type of receptor plays an important role in tumour growth, and thus the B1R receptor seems to be a good therapeutic target for novel compounds [65,66]. Furthermore, our results showed that analogues with these modifications prominently decreased the proportion of CD133+ CSCs in the spherical culture of CRC cells.

We found that the peptides with the sterically extended Aaa group at the *N*-terminal part of the sequence (BK-3, BK-4, BK-6, and BK-8) presented the most favorable anti-cancerous activity by lowering the count of CD133+ CSCs. In addition, blocking the conformational freedom on the *C*-terminal part of the compounds by placing extended, non-aromatic residues at positions 7 and 8 (BK-6 and BK-8) significantly inhibited the growth of CSC cells. These results are very promising, although they partially diverge from the previous antagonistic tests of our analogues. Our study confirms that BK analogues affect the inhibition and growth of colon cancer cells similarly to studies with glioma and lung cancer cells [30]. 

Unfortunately, in the case of NT and its analogues, the results are not so clear. Only one was promising, namely, the NT-14 peptide in which the important fragment of the sequence from residues 5 to 13 (relative to the native peptide) contain modifications at two positions. The modified positions (9 and 12) are necessary for appropriate binding to the NTRS1 receptor, which is associated with tumour growth [67]. This compound has better stability than the native peptide, which improved its potential clinical application. NT-14 also contains an additional lysine residue in the sequence, and according to literature reports, lysine is that the amino acid with significant importance in studies on CRC cell lines. Similar conclusions have been made previously by other groups [68,69]. Analysis conducted with HCT116 CRC cells proved that this extended lifetime of lysine-rich NT analogues may have a great impact on their anti-cancer effectiveness.

## 4. Materials and Methods

### 4.1. Peptide Synthesis

Peptides were synthesized via the solid-phase method using the Fmoc/*tBu* strategy as previously described [30] with minor modifications. Briefly, Fmoc-Arg-Wang Resin (GL Biochem, Shanghai, China) was used. Peptide chain extension was performed using an automated peptide synthesizer (Symphony, Gyros Protein Technologies, Warren, NJ, USA) in a reaction with a threefold excess of the respective Fmoc-amino acid in an equimolar mixture with *O*-(7-azabenzotriazol-1-yl)-*N*,*N*,*N′*,*N′*-tetramethyluronium hexafluorophosphate (HATU) and 1-hydroxy-7-azabenzotriazole (HOAt) and two equivalents of *N*-methylmorpholine (NMM). Fmoc-protected amino acids were purchased from commercial suppliers (Merck KGaA, Darmstadt, Germany; GL Biochem, Shanghai, China). Peptide cleavage was performed using a standard mixture in a 2- to 4-h reaction. Crude peptides were precipitated with cold diethyl ether, centrifuged, dried, dissolved in water, and lyophilized overnight.

### 4.2. Characterization of the Peptides

Peptides were purified using a preparative reversed-phase high-performance liquid chromatography system (RP-HPLC) (Shimadzu, Kyoto, Japan) with a Jupiter Proteo column (4 μM, 90 Å, 250 × 10 mm) (Phenomenex, Torrance, CA, USA). The purity (>95%) of the peptides was determined using an analytical RP-HPLC system (Shimadzu, Kyoto, Japan) with a Jupiter Proteo column (4 μM, 90 Å; 250 × 4.6 mm) (Phenomenex, Torrance, CA, USA) and the linear gradient of solution B in A from 5% to 95% over 30 min with a flow rate of 1 mL/min. The eluents used were as follows: A—0.1% aqueous solution of TFA and B—80% solution of acetonitrile in aqueous 0.1% TFA (*v*/*v*). The mass spectra of the peptides were recorded using a Bruker BIFLEX III and autoflex maX MALDI TOF mass spectrometers or for NT-14 only by mass spectrometry with an ESI LCMS IT TOF device (Shimadzu, Kyoto, Japan). A linear gradient solution B was applied as a mobile phase.

### 4.3. Stability Test in Human Plasma

*A.* 
*TCA procedure for peptides: BK, BK-1, BK-2, BK-5, BK-7, NT, and NT analogues.*


Peptide stability tests were performed according to a TCA stability assay based on the method used by Nguyen et al. [70]. Peptides were dissolved in water, mixed with 25% human plasma to a final concentration of 0.4 mg/mL, and incubated at 37 °C with agitation (300 rpm, Thermomixer, Eppendorf AG, Hamburg, Germany). After 0, 1, 2, 3, 4, and 24 h, 100 mL of the solution was taken and mixed with 15% TCA to obtain a final concentration of 3% (*v*/*v*). The samples were incubated in ice for 10 min and centrifuged for 10 min (12,000 rpm, Microfuge 16, Beckman Coulter, Brea, CA, USA).

*B.* 
*Formic acid/urea procedure for the peptides: BK-1, BK-2, BK-3, and BK-4.*


In the second method, peptides were dissolved in water, mixed with 25% human plasma to a final concentration 0.4 mg/mL, and incubated at 37 °C with agitation (300 rpm, Thermomixer, Eppendorf AG, Hamburg, Germany). After 0, 1, 2, 3, 4, and 24 h, 100 mL of the solution was taken and mixed with 25 μL of a mixture of 1050 µL 6 M urea, 330 µL formic acid, and 120 µL water to a final peptide concentration of 0.4 mg/mL. The samples were incubated in ice for 10 min and centrifuged for 10 min (12,000 rpm, Microfuge 16, Beckman Coulter, Brea, CA, USA). Supernatants were analyzed by RP-HPLC on column Kinetex (2.6 µm, XB-C18 100 Å 150 × 2.1 mm) (Phenomenex, Torrance, CA, USA) using a linear gradient of acetonitrile (1–80% over 20 min) with detection at 223 nm. Peptides were quantified by their peak areas relative to the initial peak areas (0 min). All stability tests were performed at least in triplicates. As control samples, peptides dissolved only in water under the same conditions were used.

### 4.4. Expansions of HCT116 and HT29 Cell Lines

We used two human colorectal cancer cell lines, namely, HT29 and HCT116 (ATCC, Manassas, VA, USA). Cells were tested for Mycoplasma contamination. The adherent form of these cell lines was expanded routinely in McCoy’s medium with 10% FBS, 1% penicillin–streptomycin, and 2 mM l-glutamine. The cells were passaged using trypsin 2–3 times/week to a point where they achieved 80% confluency. In addition, for the purposes of the current study, CRC cell lines were cultured as colonospheres (spherical 3D forms, tumourospheres) in serum-free stem cell medium (SCM). HCT116 and HT29 cell lines needed at least three passages to obtain the appropriate spherical forms necessary for inclusion in further analyses. All chemical supplements and compounds were purchased from Sigma Aldrich (Saint Louis, MO, USA). The composition of SCM was previously established by our group as the following: DMEM-F12 medium supplemented with ITS Liquid Media Complement (1×), BSA (4 mg/mL), glucose (3 mg/mL), HEPES (5 mM), l-glutamine (2 nM), progesterone (20 nM), putrescine (9.6 µg/mL), heparin (4 μg/mL), EGF (20 ng/mL), bFGF (20 ng/mL), and antibiotic–antimycotic solution (1×) [54,61].

### 4.5. Cell Treatment

HCT116 in both adherent (5 × 10^4^ cells/mL in 24-well plate) and spherical forms (8 × 10^5^/mL cells in 24-well ultra-low attachment plates) were maintained in appropriate medium. Spheres were treated for 24 h with BK and NT analogues at concentrations of 500 µg/mL and incubated at 37 °C under a humidified atmosphere of 5% CO_2_.

### 4.6. Analysis of CRC Cell Phenotype with Flow Cytometry

HCT116 cells were stained with monoclonal antibodies (BD Biosciences, San Jose, CA, USA) characteristic for some CSC-specific antigens, namely, anti-CD29-APC (clone MAR4, IgG1κ) and anti-CD44-FITC (clone C26, IgG2bκ). In addition, we used an anti-CD133/2-PE (clone 293C3, IgG2bκ) monoclonal antibody from Miltenyi Biotec (Bergisch Gladbach, Germany). Cells were incubated for 30 min in the dark, washed, and resuspended with PBS containing 1 mM EDTA for a FACS analyses, which were performed using FACS Calibur flow cytometer (BD Biosciences, San Jose, CA, USA) and BD CellQuest Pro 6.0 software. The frequency of positive cells was compared to untreated control cells. We used unstained cells to set a threshold for the positive signal.

### 4.7. Cell Death Assay (7AAD)

Following BK and NT analogue treatment of HCT116 cells, we evaluated the proportion of dead cells in our samples. For this purpose, we used a 7AAD Via Probe (BD Biosciences, San Jose, CA, USA). The cells were incubated for 30 min with 10 µm of a dye, washed with PBS, and prepared for FACS analysis. We used unstained cells to set a threshold for positive signal. The frequency of 7AAD-positive cells was compared to untreated control cells.

### 4.8. Proliferation Assay

A total of 1.5 × 104 HCT116 cells were seeded in 96-well low attachment plates in SCM, and then newly formed spheroids were treated with BK and NT analogues in duplicate. The entire experiment was repeated three times. Non-treated cells were used as a negative control. After 3 days, cell proliferation was assessed by a colorimetric CellTilter 96^®^ Aqueous One Solution Cell Proliferation Assay (Promega Corporation, Madison, WI, USA), according to the manufacturer’s instructions. Briefly, 20 μL of CellTiter 96^®^ AqueousOne Solution Reagent was added into each well of the 96-well assay plate containing the samples in 200 μL of culture medium. After 4 h of incubation, the absorbance at 490 nm was recorded using a microplate reader (Epoch, BioTek Instruments, Winooski, VT, USA).

The same test was used in the preliminary experiments with adherent HT29 cells.

### 4.9. Colonosphere Formation and Quantification

Colonospheres derived from HCT116 cells or cells isolated from CRC patients were cultured in sphere-forming media and treated with BK or NT analogues for 24 h. Then, we measured the diameter of at least 50 spheres of each experimental group with an inverted microscope (Olympus-CKX53) coupled with an Olympus SC50 digital camera (Olympus, Tokyo, Japan). Untreated cells were used as an internal control.

### 4.10. RNA Extraction, Reverse Transcription, and Real-Time PCR Experiments

HCT116 and HT29 cells in both adherent and spherical forms were maintained in appropriate medium without any BK or NT analogues and proceeded to real-time PCR to verify the presence of some expression of BK and NT receptors. Total RNA was extracted using the ExtractMe total RNA kit (Blirt, Gdańsk, Poland), according to the manufacturer’s instructions. The RNA concentrations were determined using an Epoch Microplate Spectrophotometer (BioTek, Winooski, VT, USA). Extracted RNA was reverse-transcribed and cDNA was synthesized using a Revert Aid First Strand cDNA Synthesis kit (Thermo Fisher, Walthman, MA, USA). Real-Time PCR was performed using a Step One Plus Real-Time PCR System (Life Technologies Applied Biosystems, Grand Island, NY, USA) with an AMPLIFYME SG No-ROX Mix kit (Blirt, Gdańsk, Poland). All primers were purchased from Merck (Munich, Germany). PCR was conducted with the following primers (5′-3′): BDKRB1 forward GCTGCGATCGTCTTCTTCAAC, reverse TGGTCTTGCTATCCTTGCGG; BDKRB2 forward GGCAGAGATCTACCTGGGGA, reverse GAGCCAGTCGAAGTTGTTGG; NTSR1 forward TCCTGAACACCATCATCGCC, reverse CTGAATGTGCTGTGCTCGC; SORT1 forward GGTGTTCTTCTCTTCCGTACA, reverse TTTGGCTACTACCGTCCAGAAA. The expression of the genes was normalized by the comparative –∆∆Ct method, using RPL37A as a housekeeping gene, followed by calibration (fold change) to normalized expression data of samples from the control (ratio = 1). To ensure the specificity of the PCR amplification, a dynamic melting curve analysis was performed for all reactions.

### 4.11. Statistical Analysis

Statistical analysis was performed using GraphPad Prism v 6.05 (GraphPad Software, San Diego, CA, USA). Data were subjected to a non-parametric ANOVA Kruskal–Wallis test followed by Dunn’s test as a post hoc procedure. Differences are shown as significant at *p* < 0.05. Data in figures are presented as median with min–max values.

## 5. Conclusions

Since colorectal cancer is one of the most common cancer types worldwide, efforts aimed at finding more effective treatment are urgently needed. In our work, we designed and synthesized 16 peptides, in which nine are bradykinin and its eight analogues and the remaining seven are neurotensin and its analogues. In the first stage, we tested the role of the peptides on the proliferation of two cancer cell types, and the HTC116 lines were the best. In the next step, we checked the activity of our peptides in anticancer use by an influence on the CSC spherogenic potential and phenotype of CSCs (CD133+, CD44, and CD29). The two bradykinin analogues BK-6 and BK-8 with a D-Pip at position 8 and acetylated with adamantaneacetic acid at the *N*-terminus showed the best properties in all of the studies. These peptides are the best compounds in terms of anticancer activity because they do not increase the proliferation of HTC116 CRC nor reduce their viability, and they also have no cytotoxic effect. The most crucial observation for these peptides is a reduced amount of CD133+ cancer stem cells in both forms: adherent and spherical cells. Despite these results, it seems necessary to conduct further studies to verify the full anticancer potential, specifically for these analogues. In addition, we will focus on the structure only for BK-6 and BK-8 analogues during the design of both the new tests and new sequences of these analogues.

## Figures and Tables

**Figure 1 ijms-24-09644-f001:**
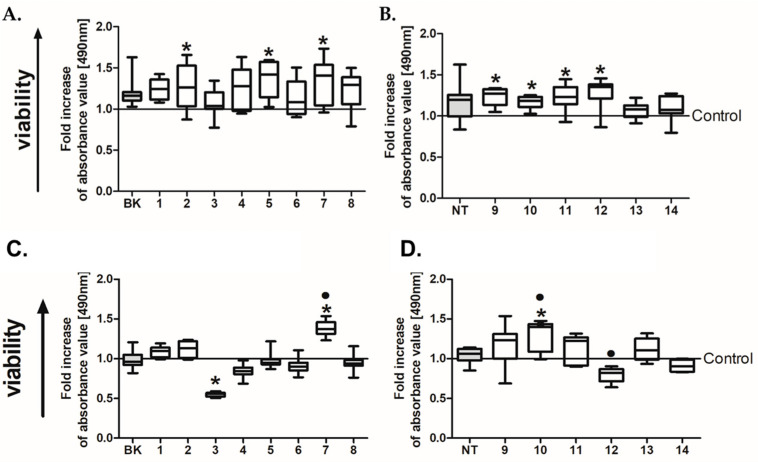
Analysis of the viability of colorectal cancer HT29 (**A**,**B**) and HCT116 (**C**,**D**) cells expanded in adherent form and treated with (**A**,**C**) bradykinin and (**B**,**D**) neurotensin analogues (500 µg/mL) for 24 h. The activity of the compounds was determined via a colorimetric assay with MTS reagent. The data are presented in relation to control untreated cells. The control value is set at the level of 1 (* *p* < 0.05 vs. untreated control cells; • *p* < 0.05 vs. BK/NT-treated cells; ANOVA Kruskal–Wallis test, *n* = 4). Data are presented as bars and whiskers representing median with min–max values.

**Figure 2 ijms-24-09644-f002:**
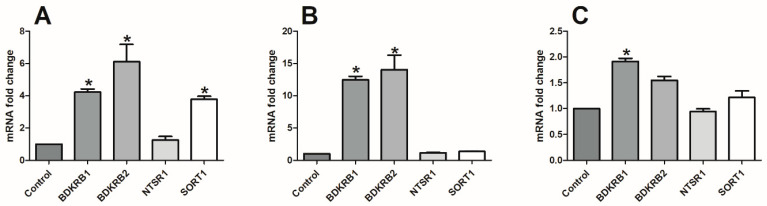
Expression of some bradykinin (B1R and B2R) and neurotensin (NTSR1 and SORT1) receptor genes in HCT116 and HT29 colorectal cancer cell lines. (**A**) Expression of BK and NT receptor genes in adherent HT29 cells relative to adherent HCT116 cells (control). (**B**) Expression of BK and NT receptor genes in spherical HCT116 cells relative to adherent counterparts (control). (**C**) Expression of BK and NT receptor genes in spherical HT29 cells relative to adherent counterparts (control). Bars and whiskers represent median with min–max values (* *p* < 0.05 vs. control cells; ANOVA Kruskal–Wallis test). Data are presented as bars and whiskers representing mean value ± SEM).

**Figure 3 ijms-24-09644-f003:**
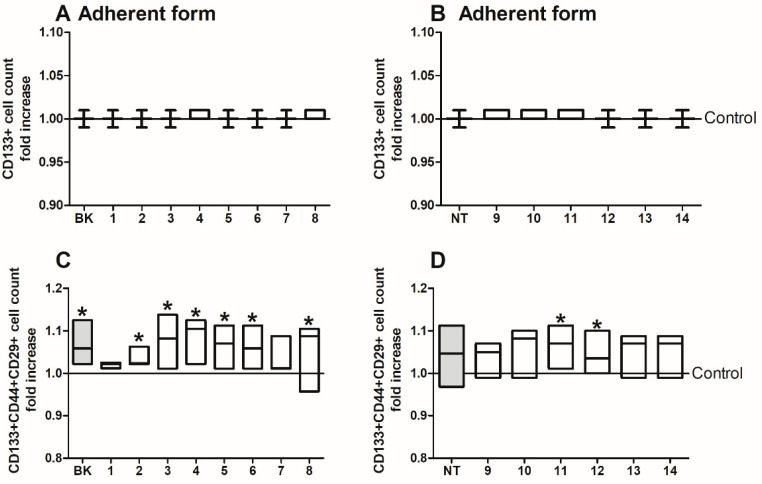
The phenotype of colorectal cancer HCT116 cell line expanded in adherent form and treated with (**A**,**C**) bradykinin and (**B**,**D**) neurotensin analogues (500 µg/mL) for 24 h. The data are presented as fold change of cells frequency with given phenotype in relation to untreated control cells. The control value was set at the level of 1. Bars and whiskers represent median with min–max values (* *p* < 0.05 vs. untreated control cells; ANOVA Kruskal–Wallis test, *n* = 5). (**C**,**D**) The cytotoxicity of the proteins was determined via colorimetric assay with MTS reagent, with the results shown relative to untreated control cells (* *p* < 0.05 vs. untreated control cells; ANOVA Kruskal–Wallis test, *n* = 4). Data presented as bars and whiskers representing the median with min–max values).

**Figure 4 ijms-24-09644-f004:**
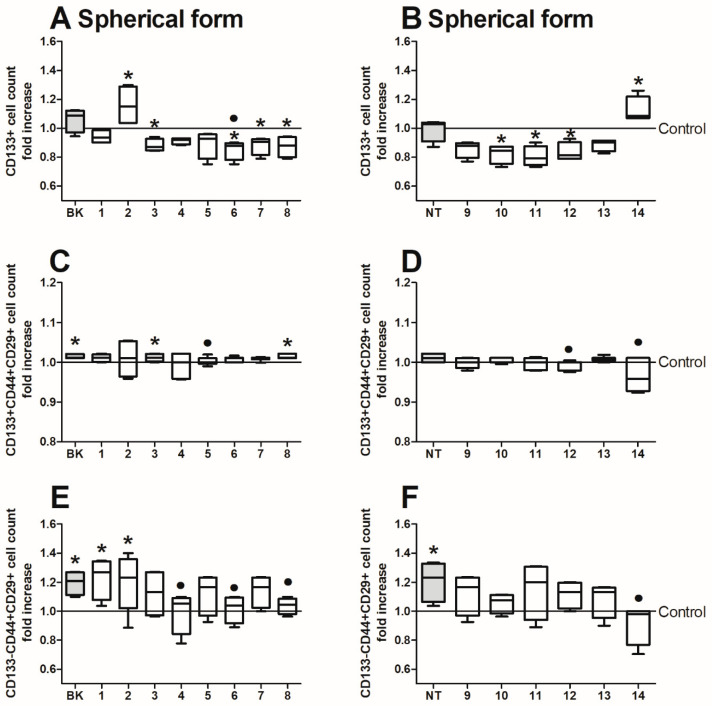
The phenotype of the colorectal cancer HCT116 cell line expanded in the form of colonospheres and treated with (**A**,**C**,**E**) bradykinin and (**B**,**D**,**F**) neurotensin analogues (500 µg/mL) for 24 h. The data are presented as fold change of frequency of cells with given phenotype in relation to untreated control cells. The control value is set at a level of 1. Bars and whiskers represent the median with min–max values (* *p* < 0.05 vs. untreated control cells; • *p* < 0.05 vs. BK treated cells; ANOVA Kruskal–Wallis test, *n* = 5).

**Figure 5 ijms-24-09644-f005:**
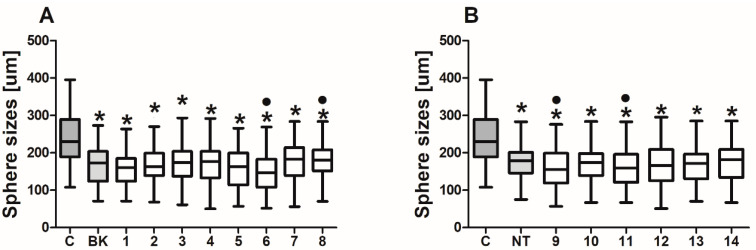
Sizes of colonospheres. Colonospheres were derived from the HCT116 colorectal cancer cell line and treated with (**A**) bradykinin or (**B**) neurotensin peptides (500 µg/mL) for 24 h. The diameter of at least 80 spheres of each experimental group was measured with an inverted microscope and a digital camera. Bars and whiskers represent the median with min–max values (* *p* < 0.05 vs. untreated control cells (C); • *p* < 0.05 vs. native BK/NT-treated cells; ANOVA Kruskal–Wallis test).

**Figure 6 ijms-24-09644-f006:**
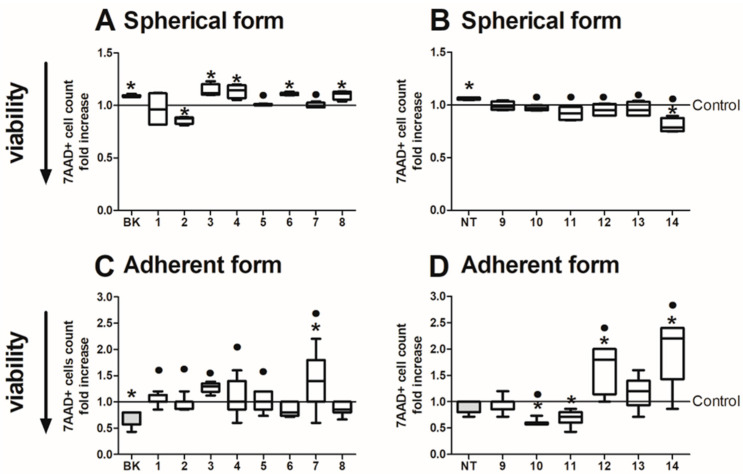
Analysis of the viability of colorectal cancer HCT116 cells expanded in spherical (**A**,**B**) or adherent (**C**,**D**) form and treated with bradykinin (**A**,**C**) and neurotensin analogues (**B**,**D**) (500 µg/mL) for 24 h. Cells were incubated with 7-AAD and analyzed by flow cytometry. The *y*-axis presents the fold increase in the frequency of 7-AAD+ cells compared to untreated control cells. (* *p* < 0.05 vs. untreated control cells; • *p* < 0.05 vs. BK/NT-treated cells; ANOVA Kruskal–Wallis test, *n* = 4). Data presented as bars and whiskers representing the median with min–max values).

**Table 1 ijms-24-09644-t001:** Sequences and characterization of BK and its peptide analogues. Aaa—[(*3S*,*5S*,*7S*)-adamantan-1-yl]acetic acid; Acc—1-aminocyclohexane-1-carboxylic acid; Apc—1-aminocyclopentyl-1-carboxylic acid; Thi—thienylalanine; L-Pip—(L)-piperidine-2-carboxylic acid; and Hyp—hydroxyproline.

Peptide ID	Sequence											MW
	0′	0	1	2	3	4	5	6	7	8	9	
BK			Arg	Pro	Pro	Gly	Phe	Ser	Pro	Phe	Arg	1060.66
BK-1		d-Arg	Arg	Pro	Hyp	Gly	Thi	Ser	Apc	Thi	Arg	1258.59
BK-2		d-Arg	Arg	Pro	Hyp	Gly	Thi	Ser	d-Phe	Apc	Arg	1252.76
BK-3	Aaa	d-Arg	Arg	Pro	Hyp	Gly	Thi	Ser	Acc	Thi	Arg	1448.87
BK-4	Aaa	d-Arg	Arg	Pro	Hyp	Gly	Thi	Ser	d-Phe	Acc	Arg	1442.89
BK-5		d-Arg	Arg	Pro	Hyp	Gly	Thi	Ser	d-Phe	l-Pip	Arg	1252.65
BK-6	Aaa	d-Arg	Arg	Pro	Hyp	Gly	Thi	Ser	d-Pip	l-Pip	Arg	1428.75
BK-7		d-Arg	Arg	Pro	Hyp	Gly	Thi	Ser	d-Pip	Thi	Arg	1258.58
BK-8	Aaa	d-Arg	Arg	Pro	Hyp	Gly	Thi	Ser	d-Pip	Thi	Arg	1434.70

**Table 2 ijms-24-09644-t002:** Sequences and characterization of NT and its peptides analogues. Tle—L-*tert*-leucine.

Peptide ID	Sequence													MW
	1	2	3	4	5	6	7	8	9	10	11	12	13	
NT	Glu	Leu	Tyr	Glu	Asn	Lys	Pro	Arg	Arg	Pro	Tyr	Ile	Leu	1690.23
NT-9								Arg	Arg	Pro	Tyr	Ile	Leu	817.46
NT-10								Arg	Arg	Pro	Ala	Ile	Leu	725.43
NT-11								Arg	Arg	Pro	Tyr	Ala	Leu	775.41
NT-12					Pro	Glu	Gly	Arg	Lys	Pro	Tyr	Tle	Leu	1072.58
NT-13					Pro	Glu	Gly	Lys	Arg	Pro	Tyr	Tle	Leu	1072.58
NT-14					Pro	Glu	Gly	Lys	Lys	Pro	Tyr	Tle	Leu	1044.58

## Data Availability

The datasets supporting the conclusions and description of a complete protocol can be found within the manuscript and its additional files. The datasets used and/or analyzed during the current study are available on request from the corresponding author.

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
