# Peer review of "Bradykinin and Neurotensin Analogues as Potential Compounds in Colon Cancer Therapy"

_ijms, 2023, doi:10.3390/ijms24119644_

Round 1
Reviewer 1 Report
The manuscript “The bradykinin and neurotensin analogues as potential compounds in anti-colon cancer therapy” by M. Szaryńska et al. describes a pattern of modified peptides of bradykinin (BK) and neurotensin (NT) and analyzed their antitumor activity using CRC cell lines cultured in adherent or spherical mode. A deep characterization of CD133+ cancer stem cells (CSCs) is performed on cell treated with the BK and NT derived peptides.
Some aspects have to be revised.
The title: “anti” (anti-colon cancer therapy) has to be removed from the title.
It is not clear from the fig S16 that BK7 is completely nearly digested. Clarifying this aspect.
The results of experiments performed on HCT116 cells are not reported (paragraph 2.3, line 198), which are crucial because from there the authors decide to focus on this cell line.
Experiments not reported in the manuscript are described in the discussion (lines 325-327).
Author Response
First of all, we thank the Reviewer for time and effort spent on checking our manuscript “The bradykinin and neurotensin analogues as potential compounds in anti-colon cancer therapy”. Each comment has been carefully considered point by point and responded.
All changes in the revised version of our manuscript have been made in the change tracking mode.
- The title: “anti” (anti-colon cancer therapy) has to be removed from the title.
Reply: Thank You for this remark. We changed the title.
- It is not clear from the fig S16 that BK7 is completely nearly digested. Clarifying this aspect.
Replay: This is our mistake, because when describing the results, we mistakenly analyzed a file containing also other analogs. We apologize for this mistake. We have corrected the description in the manuscript to be consistent with the results as below:
Bradykinin as a native peptide and as a control is a relatively stable compound. After 4 h of the incubation no significant degradation was observed (96% of the peptide concentration) while after 24 h in the test sample about 50% of the peptide remained. A similar situation was observed for the peptides BK-1, and BK7. Slightly higher stability, after 24 hours in the test sample was 70% of the peptide, had analogue BK-8. The most stable were analogues: BK-2, BK-4 and BK-5. The last one and the least stable was BK-3. Its final concentration was about 40% of the initial concentration value. (Supplementary Data, Figure S17)
- The results of experiments performed on HCT116 cells are not reported (paragraph 2.3, line 198), which are crucial because from there the authors decide to focus on this cell line.
Replay: We are very sorry for this confusing issue. The decision concerning the CRC cell line which we used during our study was based on the proliferation test (mentioned in method paragraph in point 4.8). The initial study was conducted on the adherent HT29 and HCT116 cell lines. The results concerning HT29 cell line is presented on the Figure 1 whereas results concerning HCT116 – on the Figure 6E and F).
We added the information clarifying this issue into the text.
Additionally, we added the information that this method was used for HT29 cells in preliminary experiments into 4.8 Methods subparagraph.
- Experiments not reported in the manuscript are described in the discussion (lines 325-327).
Replay: In the discussion, we referred to the of the experiment from our studies conducted iand discus in the publication and to our previous studies involving these compounds. We have added references when we compare our results to the studies from the literature, so that it is clear. We have clarified this and have added the relevant references.
Reviewer 2 Report
In this manuscript, the authors report ‘The bradykinin and neurotensin analogues as potential compounds in anti-colon cancer therapy’.
Before I make my comments, I must note that this manuscript has not been well reviewed by the authors, as it contains many typos, and they should send a revised and reviewed version below. For a journal like IJMS, a manuscript cannot be written like the first version of a student paper. My specific comments can be found below:
1. Abstract
Authors already in the Abstract write in vivo instead in vivo. Please, check your tip. Last sentence: explain what further analyses are planned for the future.
2. Introduction
In the Introduction section authors do not care about spaces (line 27).
Line 56, 61: Italic for N- and C-
Line 59, 60,: Descriptors like L- and D- should be written in smaller font; for example: if Manuscript is written in font 12, L- and D- should be 10.
Line 81, 96: in vivo in Italic form
3. Results
Since authors describe synthesis and talk about β-turn (line 129, beta shoul be written in Symbol font) ,they should provide some experiment or reference which prove the formation of β-turn (NMR-, CD- experiments). If they do not have results that prove it, they should not presume that β-turn is formed.
Line 136: Descriptors like L- and D- should be written in smaller font
Line 138: Italic for N-
Lines 143, 144: Please, reformulate the sentence: 'Since the active site of this compound is located between residues 8-13, why we have made modifications in this fragment. '
It is not intelligible what authors wanted to say.
Line 155: please reference your experience by putting appropriate references.
Line 163: Descriptors like L- and D- should be written in smaller font
Table 1: L-Pip or L-pip?
4. Discussion
In vitro, N-terminal, L-, D-, spaces….
5. Materials and Methods
Line 388: tBu, tert should be in Italic
Line 389: 83 is what?
6. References
The references are not properly written, authors should check MDPI citing way.
Supplementary data:
The chromatogram RP-HPLC (showing the purity of the peptides) should not be trimmed: The retention time must be indicated in the graph, as well as the table below the chromatogram showing the percentage of the peak area of each peak, so that the reader can see that the area of the peak of the synthesized peptide is really > 95%. As an example, Figure S4: I am not sure if the peak area of BK -3 peptide is really > 95%.
Figure S3: [M] is not good calculated? Authors need to repeat this experiment- Molar mass differs by six protons.
Figure S 11: Maldi-TOF graph is cropped, why? Calculated mass: 1434.56; found 1434.70, graph ends at cca 1000.
For the reasons stated above, I suggest that the authors revise the manuscript in the spirit of these comments.
-
Author Response
In this manuscript, the authors report ‘The bradykinin and neurotensin analogues as potential compounds in anti-colon cancer therapy’.
Before I make my comments, I must note that this manuscript has not been well reviewed by the authors, as it contains many typos, and they should send a revised and reviewed version below. For a journal like IJMS, a manuscript cannot be written like the first version of a student paper. My specific comments can be found below:
Replay: First of all, we thank the Reviewer for time and effort spent on checking our manuscript “The bradykinin and neurotensin analogues as potential compounds in anti-colon cancer therapy”. Each comment has been carefully considered point by point and responded.
All changes in the revised version of our manuscript have been made in the change tracking mode.
We apologize for our English. We made a language proofreading of the manuscript to improve its quality and to avoid any misunderstandings.
1.Abstract
Authors already in the Abstract write in vivo instead in vivo. Please, check your tip. Last sentence: explain what further analyses are planned for the future.
Replay: Did You mean that we should write “in vitro” instead “in vivo”? Did we understand Your remark correctly? If Yes, we wanted to highlight the fact that 3D model of cancer cells culture more resembles the solid tumor structure.
Or we should use italic when we wrote "in vivo"? Beacuse in this part of Your Review Report we don't see any italic suggestion. If Yes, we changed italic formula "in vivo" and "in vitro" in the whole text.
Future plans: The results presented in our current manuscript were aimed to verify which chemical modifications of bradykinin and neurotensin are the most promising for their anti-cancerous potential. The most advantageous peptides will be included into next experiments assessing the molecular changes (eg. the expression of receptors, target proteins, markers of apoptosis, senescence, cell cycle arrest) induced after the treatment of cancer cells. Additionally, we plan to analyze the modulatory features of BK and NT analogues with different leukocytes’ populations.
- Introduction
In the Introduction section authors do not care about spaces (line 27).
Line 56, 61: Italic for N- and C- Checked and done
Line 59, 60,: Descriptors like L- and D- should be written in smaller font; for example: if Manuscript is written in font 12, L- and D- should be 10. - Checked and done
Line 81, 96: in vivo in Italic form - Checked and done
Replay: We apologize for this situation but we would like to clarify that in our version of the manuscript many of the mentioned mistakes not exist. We observed some extra spaces and we changed these. After download manuscript from the Editorial Office, really it looked as the reviewer describes. This is not our fault but we changed once again all mistakes and we have hope that now it will look correct. If You want we can sent our PDF file, to show how it really was.
- Results
Since authors describe synthesis and talk about β-turn (line 129, beta shoul be written in Symbol font) ,they should provide some experiment or reference which prove the formation of β-turn (NMR-, CD- experiments). If they do not have results that prove it, they should not presume that β-turn is formed.
Line 136: Descriptors like L- and D- should be written in smaller font - Checked and done
Line 138: Italic for N- Checked and done
It is not intelligible what authors wanted to say.
Line 155: please reference your experience by putting appropriate references.
Line 163: Descriptors like L- and D- should be written in smaller font Checked and done
Table 1: L-Pip or L-pip?
Replay: Of course, we agree with the Reviewer that when discussing β-turns, we should do more research such as CD or NMR studies. Nevertheless, the discussion was based on the research of other scientists, which work with bradykinin analogues. They checked the effect of such modifications for the conformation of peptides for many years. Due to the fact, that they are respected and well-known researchers, we did not challenge this theory in any way. However, if the Reviewer wishes to check this using a CD techniques, we of course will do this experiment.
We have corrected, we hope, all editing errors, which, as we wrote above, are not our fault. We suppose this is the result of working with the file on different operating systems. We know from experience that this is often the case.
Line 143: We changed for that one: As the active site of this compound is located between residues 8-13, we made modifications in this fragment.
Line 155 the references were added.
- Discussion
In vitro, N-terminal, L-, D-, spaces….
Replay: We apologize for that and we checked once again our manuscript.
- Materials and Methods
Line 388: tBu, tert should be in Italic – it is done
Line 389: 83 is what? –
We apologize it is our editorial mistake, it should be reference for the stability techniques, we add correct.
- References
The references are not properly written, authors should check MDPI citing way.
Replay: During the writing of this manuscript, we used the Mendeley program. We trusted, that the selection of the pattern of appropriate citation in the program would be in accordance with IJMS (this is what we selected in the program). Most likely, during the transfer to the template file there was bad formatting, which we did not notice. In the review manuscript we made changes manually and we have hope that now it is as it should be .
Supplementary data:
The chromatogram RP-HPLC (showing the purity of the peptides) should not be trimmed: The retention time must be indicated in the graph, as well as the table below the chromatogram showing the percentage of the peak area of each peak, so that the reader can see that the area of the peak of the synthesized peptide is really > 95%. As an example, Figure S4: I am not sure if the peak area of BK -3 peptide is really > 95%.
Figure S3: [M] is not good calculated? Authors need to repeat this experiment- Molar mass differs by six protons.
Figure S 11: Maldi-TOF graph is cropped, why? Calculated mass: 1434.56; found 1434.70, graph ends at cca 1000.
Thank the Reviewer for these comments. Regarding the first one, we have registered the new chromatograms once again for all the peptides and showed the retention time on the graphs. We did not showing the percentage of the peak area of each peak because in the characterization of peptides we never did that. We didn't encounter something like that in any publication related to peptides.
Figure S3: You have right it is incorrect data in the file. We calculate once again molecular mass for this peptide (BK-2) using ChemSketch and it equal 1252,76. We did new mass spectra for the peptide and now it is correct. We calculated molecular mass for all peptides and verified in the main file and SD.
Figure S11: the mass of the NT-14 is 1044.24. In the mass spectra from ESI iI ToF the pseudomolecular ions (after fragmentation) are observed such as: [M+2]2+ its mean that the MW of the petide is 1044.24. Because if we observed ion 522.79 we can calculate: (522,79*2)+2 and equal 1047, 58. To confirm this we can send a spectrum after deconvolution. In the main fail of the publication (table 2) the mass of this peptide was correct.
Reviewer 3 Report
In this manuscript, Emilia and co-workers designed, synthesized, and tested the efficiency of the bradykinin (BK) and neurotensin peptide derivatives against colon cancer. These analogs comprised a few changes within the previously known sequences and were found to have a low impact on the viability of colorectal cancer cells. Overall, the authors attempted to improve the efficiency, however, the authors didn't talk about the rationality of these peptide modifications.
In my opinion, this manuscript needs a lot of improvements before its final publication.
1. Can authors include any computational data (eg., SAR) for the rational design of peptide sequences?
2. Did the authors check other concentrations rather than 500 µg/mL, if so please provide those details in the manuscript.
3. It seems that the masses of BK-2 and NT-9 peptides were mimatching, can authors retake the data?
4. Did the authors consider about the stapling or the cyclization of the peptide sequences?
5. Please correct the typo's throughout the manuscript.
Author Response
In this manuscript, Emilia and co-workers designed, synthesized, and tested the efficiency of the bradykinin (BK) and neurotensin peptide derivatives against colon cancer. These analogs comprised a few changes within the previously known sequences and were found to have a low impact on the viability of colorectal cancer cells. Overall, the authors attempted to improve the efficiency, however, the authors didn't talk about the rationality of these peptide modifications.
In my opinion, this manuscript needs a lot of improvements before its final publication.
First of all, we thank the Reviewer for time and effort spent on checking our manuscript “The bradykinin and neurotensin analogues as potential compounds in anti-colon cancer therapy”. Each comment has been carefully considered point by point and responded.
All changes in the revised version of our manuscript have been made in the change tracking mode.
We somewhat agree with the Reviewer, that we do not discuss the rationality of the selected peptides in great depth. This is because the conducted researches were more like preliminary and screening studies. In the future, in fact now we are working on a more extensive structural analysis of these peptides as well as more studies towards cancer therapy.
- Can authors include any computational data (eg., SAR) for the rational design of peptide sequences?
Replay: No we can’t. For the study we took peptides that were designed in our laboratory some time ago in the context of completely different research. Their results prompted us to test them also in the context of anti-cancer properties.
- Did the authors check other concentrations rather than 500 µg/mL, if so please provide those details in the manuscript.
Replay: In our preliminary data we conducted the cytotoxic tests with the use of tested analogues in the concentration ranging from 500 µg/ml to 1.7 µg/ml. Since our compounds induced the proliferation of HT29 cells we decided to present these results in simpler form to avoid overwhelming of our manuscript.
- It seems that the masses of BK-2 and NT-9 peptides were mimatching, can authors retake the data?
Replay: We rechecked both peptides. The calculated mass for BK-2 is 1252.46 and for NT-9 is 816.99. We recorded the new mass spectra for BK-2 and changed to the correct files. For NT-9 peptide the mass spectra is correct only the calculation mass was wrong. Sequence of this peptide is RRPYIL, and MW is 816.99. In the manuscript (table 2) the correct mass is available.
- Did the authors consider about the stapling or the cyclization of the peptide sequences?,
Replay: Yes, because there are known cases in the literature that dimers of peptides of this type can actually have better activity. Therefore, we are considering to design dimers or even cyclic compounds in the future.
- Please correct the typo's throughout the manuscript.
Replay: The Typo’s have been corrected.
Round 2
Reviewer 1 Report
The manusript is now ready for publication
Author Response
Thank You very much!
Reviewer 3 Report
The authors undertook the consideration of the reviewer's comments and addressed them in an appropriate manner. However, the following changes must be done before its final publication.
1. Please replace pure products with purified products.
2. Please correct all technical errors throughout the manuscript.
About my comment 2:
The manuscript is poorly written, which made me difficult to understand the science presented by the authors. I'm worried that I might have missed some valuable scientific conclusions. Therefore, I would recommend polishing the English language before its final publication.
Author Response
Than You very much once again
1. Please replace pure products with purified products.
Replay: Done
2. Please correct all technical errors throughout the manuscript.
About my comment 2:
The manuscript is poorly written, which made me difficult to understand the science presented by the authors. I'm worried that I might have missed some valuable scientific conclusions. Therefore, I would recommend polishing the English language before its final publication.
Replay:
Thank you very much for the clarification Your review.
We would like to clarify some acpects.
Our publication has been linguistically corrected by a native speaker in chemistry and biology area, with whom we have been working for years. We are therefore a little surprised by the Reviewer's opinion and request for new correction. We are aware that the results of our research could be inconclusive and not spectacular. If they were obviously, it could be more easy for us to describe and discuss them. Unfortunately, , in our research, depending on the technique chosen to determine the anticancer potential of our compounds, the results were not consistent with what was reported in the literature, little, they did not always overlap. Therefore, we worked in a multifaceted manner, with different types of cell lines (HT29, HTC116, adherent spherical etc) to obtain the most reliable and complete picture for compounds in whole experiments. Perhaps, the lack of clear sentences in the text has just given the impression that the results are somewhat unclear. For which we are very sorry. We have tried to correct this throughout the publication and hope that this version of the manuscript is more clear.